# Multiple kinases inhibit origin licensing and helicase activation to ensure reductive cell division during meiosis

David V Phizicky[1,2], Luke E Berchowitz[3], Stephen P Bell[1,2]*

[1]Department of Biology, Massachusetts Institute of Technology, Cambridge, United States; [2]Howard Hughes Medical Institute, Maryland, United States; [3]Department of Genetics and Development, Columbia University Medical Center, New York, United States

**Abstract** Meiotic cells undergo a single round of DNA replication followed by two rounds of chromosome segregation (the meiotic divisions) to produce haploid gametes. Both DNA replication and chromosome segregation are similarly regulated by CDK oscillations in mitotic cells. Yet how these two events are uncoupled between the meiotic divisions is unclear. Using *Saccharomyces cerevisiae*, we show that meiotic cells inhibit both helicase loading and helicase activation to prevent DNA replication between the meiotic divisions. CDK and the meiosis–specific kinase Ime2 cooperatively inhibit helicase loading, and their simultaneous inhibition allows inappropriate helicase reloading. Further analysis uncovered two previously unknown mechanisms by which Ime2 inhibits helicase loading. Finally, we show that CDK and the polo–like kinase Cdc5 trigger degradation of Sld2, an essential helicase–activation protein. Together, our data demonstrate that multiple kinases inhibit both helicase loading and activation between the meiotic divisions, thereby ensuring reductive cell division.

DOI: https://doi.org/10.7554/eLife.33309.001

*For correspondence: spbell@mit.edu

## Introduction

The production of haploid gametes is required for sexual reproduction. These gametes are produced by meiosis, a specialized cell division program during which a single round of DNA replication is followed by two rounds of chromosome segregation (the meiotic divisions), Meiosis I (MI) and Meiosis II (MII). In contrast, mitotically–dividing cells maintain their ploidy by strictly alternating rounds of DNA replication and chromosome segregation. The lack of DNA replication between MI and MII is essential for the reduction in ploidy inherent to meiosis, but it is unclear how the meiotic program differs from mitosis to allow for two sequential chromosome segregation events without an intervening S phase.

In mitotic cells, both DNA replication and chromosome segregation require cyclin-dependent kinase (CDK) activity to oscillate during the cell cycle. A low–CDK state during G1 phase allows both events to initiate, and a high–CDK state is required for their completion. During meiosis, the CDK–oscillation dependence of both events presents a unique problem between MI and MII, a period known as the MI–MII transition (*Figure 1A*). After MI has been completed, CDK activity decreases, and then increases again upon entry into MII (*Carlile and Amon, 2008*). This oscillation is required for multiple essential chromosome–segregation events, including duplication of the spindle pole body (SPB, the yeast centrosome) (*Buonomo et al., 2003*; *Fox et al., 2017*; *Marston et al., 2003*). However, the DNA replication program must remain inhibited between MI and MII to achieve the hallmark of meiosis, reductive cell division. Given that an oscillation of CDK activity is sufficient for re–replication of the entire genome in mitotic cells (*Dahmann et al., 1995*), it is not fully understood

**Figure 1.** Mcm2–7 loading onto replication origins is inhibited during the MI–MII transition. (**A**) The DNA replication program and chromosome segregation program are uncoupled during the MI–MII transition. Relative CDK activity at various stages of the meiotic cell cycle are shown (*Carlile and Amon, 2008*). The dashed boxes highlight the oscillations of low-to-high CDK activity during meiosis, and show the discrepancy between CDK-regulation of SPB duplication and DNA replication. See text for details. (**B**) ORC is bound to origins of replication throughout the meiotic divisions. The strain yDP71 was put through meiosis. ChIP–qPCR was used to detect ORC binding at the early–firing origin *ARS1* (top graph, dark blue), the late–firing origin *ARS1413* (top graph, light blue), and at the re–replication prone origins ARS305 (bottom graph, dark blue) and ARS418 (bottom graph, light blue). The time after transfer into sporulation medium and the associated meiotic stages are indicated below each lane. For cell–cycle stage quantification for this experiment, see *Figure 1—figure supplement 1A*. The peak % of input DNA immunoprecipitated (set to arbitrary unit (A.U.)

*Figure 1 continued on next page*

*Figure 1 continued*

=1.0) was 9.1% for ARS1, 2.2% for ARS1413, 46.7% for ARS305, and 10.9% for ARS418. (C) Mcm2–7 is bound to origins of replication in G1 phase but does not reassociate with origins during or between the meiotic divisions. The strain yDP71 was put through meiosis. ChIP–qPCR was used to detect Mcm2–7 binding at the early–firing origin *ARS1* (top graph, red), the late–firing origin *ARS1413* (top graph, orange), and at the re–replication prone origins ARS305 (bottom graph, red) and ARS418 (bottom graph, orange). The time after transfer into sporulation medium and the associated meiotic stages are indicated below each lane. For cell–cycle stage quantification for this experiment, see *Figure 1—figure supplement 1B*. The peak % of input DNA immunoprecipitated (set to A.U. = 1.0) was 7.4% for ARS1, 5.0% for ARS1413, 35% for ARS305, and 19.5% for ARS418.

DOI: https://doi.org/10.7554/eLife.33309.002

The following source data and figure supplements are available for figure 1:

**Source data 1.** Raw values used for the quantification of *Figure 1B and C*.
DOI: https://doi.org/10.7554/eLife.33309.006

**Figure supplement 1.** Cells from *Figure 1* proceeded synchronously through meiosis.
DOI: https://doi.org/10.7554/eLife.33309.003

**Figure supplement 2.** Both ORC and Mcm2-7 associate specifically with origins of replication compared to non-origin DNA.
DOI: https://doi.org/10.7554/eLife.33309.004

**Figure supplement 3.** Mcm2–7, Cdt1, and Orc1 proteins are present throughout meiosis.
DOI: https://doi.org/10.7554/eLife.33309.005

how meiotic cells reset the chromosome segregation program while retaining inhibition of the DNA replication program.

Mitotic cells use oscillations of CDK activity to ensure that the genome is replicated exactly once per cell division. During G1 phase, low CDK activity allows for the Mcm2–7 complex, the core enzyme of the replicative helicase, to be loaded onto origins of replication in an inactive state. This event, known as origin licensing or helicase loading, cannot occur in the presence of high CDK activity and requires the cooperative action of three proteins: Cdc6, Cdt1, and the Origin Recognition Complex (ORC) (*Evrin et al., 2009*; *Remus et al., 2009*). Upon S–phase entry, S–CDK (CDK bound to S-phase cyclins Clb5/6) is activated and impacts DNA replication in two ways. First, S–CDK phosphorylates two essential proteins, Sld2 and Sld3, that subsequently promote helicase activation, replisome assembly, and chromosome duplication (*Masumoto et al., 2002*; *Tanaka et al., 2007*; *Zegerman and Diffley, 2007*). Second, both S–CDK and M-CDK (CDK bound to mitotic cyclins Clb1-4) inhibit new helicase loading during S, G2, and M phases. These kinases directly phosphorylate Cdc6, Mcm3, and ORC to trigger the proteolytic degradation of Cdc6, the nuclear export of Mcm2–7–Cdt1, and inhibition of ORC helicase–loading activity, respectively (*Calzada et al., 2000*; *Chen and Bell, 2011*; *Drury et al., 2000*; *Labib et al., 1999*; *Nguyen et al., 2000*).

CDK oscillations also ensure that chromosome segregation occurs once per mitotic cell cycle (*Winey and Bloom, 2012*). At the end of G1 phase, G1–CDK (CDK bound to G1 cyclins Cln1-3) is required for duplication of the SPB (*Jaspersen et al., 2004*). Later in the cell cycle, S–CDK and M–CDK prevent re–duplication of SPBs, and M-CDK is essential for the assembly of metaphase spindles (*Avena et al., 2014*; *Elserafy et al., 2014*; *Haase et al., 2001*). Finally, downregulation of CDK activity is required for anaphase spindle disassembly upon completion of chromosome segregation (*Shirayama et al., 1999*; *Thornton and Toczyski, 2003*; *Wäsch and Cross, 2002*). From this point forward, we will refer only to total CDK activity without specifying G1–, S–, or M–CDK, as the events we will be discussing are similarly regulated by all three kinases.

Two models have been proposed to explain how meiotic cells uncouple DNA replication and chromosome segregation during the MI–MII transition. The CDK–balance model suggests that partially inactivating CDK is sufficient to reset the chromosome segregation program while still inhibiting Mcm2–7 loading and replication initiation (*Iwabuchi et al., 2000*). In contrast, the alternative–kinase model suggests that a second kinase inhibits Mcm2–7 loading during the MI–MII transition, allowing the oscillation of CDK activity to reset the chromosome segregation program without resetting the DNA replication program. Ime2, a yeast meiosis–specific kinase that is evolutionarily related to CDK (*Krylov et al., 2003*), has been proposed to have this role (*Holt et al., 2007*).

We set out to systematically address how DNA replication is inhibited between the meiotic divisions using a combination of in vivo and in vitro approaches. We found that Mcm2–7 loading is the earliest inhibited step of replisome assembly during the MI–MII transition, and that this inhibition can be bypassed by simultaneous inhibition of CDK and Ime2. Furthermore, we identified two

previously uncharacterized mechanisms used by Ime2 to inhibit origin licensing. First, Ime2 phosphorylation of Mcm2–7 directly inhibited its ability to be loaded onto replication origins. Second, we found that Ime2 and CDK cooperate to repress expression of Cdc6. In addition to the inhibition of helicase loading, meiotic cells promote degradation of Sld2, an essential helicase–activation protein, using phosphorylation sites for CDK and the polo–like kinase Cdc5. Together, these data show that meiotic cells use multiple kinases to inhibit both Mcm2–7 loading and activation, ensuring that MI and MII occur sequentially without an intervening S–phase.

## Results

### Mcm2–7 loading is the first inhibited step of DNA replication initiation

To address how DNA replication is inhibited between MI and MII, we sought to identify the earliest inhibited step of replisome assembly. During replication initiation, the first proteins to stably associate with replication origins are ORC followed by the Mcm2–7 complex (*Bell and Labib, 2016*). To analyze ORC and Mcm2–7 binding to origins in populations of cells, it was necessary for us to obtain synchronized cultures of cells undergoing meiosis. To this end, we used a previously–described block–release procedure (*Benjamin et al., 2003*; *Carlile and Amon, 2008*). This method allows meiotic cells to proceed through G1 and S phase but arrest in Prophase I (hereafter referred to as G2 phase). Subsequent release from this cell cycle arrest results in cells progressing synchronously through Metaphase I, Anaphase I, Metaphase II, and Anaphase II with >90% of cells completing meiosis (*Figure 1—figure supplement 1*).

Using ChIP–qPCR, we found that ORC was bound to a representative early–firing (*ARS1*) and late–firing (*ARS1413*) replication origin throughout both meiotic divisions (*Figure 1B*, top). ORC binding to DNA is therefore not limiting for replication initiation during meiosis. In contrast, although Mcm2–7 was present at origins during pre–meiotic G1 phase, this complex did not associate with either *ARS1* or *ARS1413* throughout MI and MII (*Figure 1C*, top). We also tested two additional origins, *ARS305* and *ARS418*, that are prone to Mcm2–7 reloading and DNA re–replication in mitotic cells (*Green et al., 2006*; *Nguyen et al., 2001*; *Tanny et al., 2006*). As with the other origins tested, ORC bound to these origins throughout the meiotic divisions but Mcm2–7 loading was not observed after pre–meiotic G1 phase (*Figure 1B and C*, bottom). Both ORC and Mcm2-7 associated specifically with origin DNA sequences compared to non-origin DNA (*Figure 1—figure supplement 2*). The lack of Mcm2–7 binding to origins was not due to the absence of Mcm2–7 proteins or the essential helicase–loading protein Cdt1 (*Figure 1—figure supplement 3*). Thus, Mcm2–7 loading onto origins of replication is inhibited during the MI–MII transition.

### CDK-dependent mechanisms inhibiting Mcm2–7 loading are weakened during the MI–MII transition

We considered two reasons that Mcm2–7 complexes were not reloaded upon decreased CDK activity during the MI–MII transition: (1) the known CDK-dependent mechanisms remain active enough to completely inhibit helicase loading (the CDK-balance model); or (2) meiosis–specific mechanisms inhibit helicase loading while CDK activity is reduced (the alternative-kinase model). To test the first possibility, we asked whether any of the CDK–dependent mechanisms preventing Mcm2–7 loading in mitotic cells were weakened during the MI–MII transition. In mitotically dividing cells, CDK inhibits helicase loading by three mechanisms: inhibition of ORC function, Cdc6 protein degradation, and Mcm2–7 nuclear export (*Arias and Walter, 2007*). We found that at least two of these mechanisms were transiently weakened during the MI–MII transition.

Phosphorylation of Orc2 and Orc6 by CDK prevents ORC from facilitating helicase loading (*Chen and Bell, 2011*; *Nguyen et al., 2001*). Consistent with robust helicase loading during pre–meiotic G1 phase (*Figure 1C*), Orc2 and Orc6 were not phosphorylated during G1 when CDK is inactive (*Figure 2A and B*) (*Carlile and Amon, 2008*). In contrast, both subunits were phosphorylated throughout MI when CDK is highly active. Interestingly, we observed partial de–phosphorylation of both Orc2 (*Figure 2A* lane 9) and Orc6 (*Figure 2B* lane 6) at the MI–MII transition, before returning to a fully phosphorylated state that persisted until the end of MII.

Unlike ORC inhibition, CDK–phosphorylation of Cdc6 targets it for proteolytic degradation (*Calzada et al., 2000*; *Drury et al., 2000*) and CDK is also partially responsible for repression of

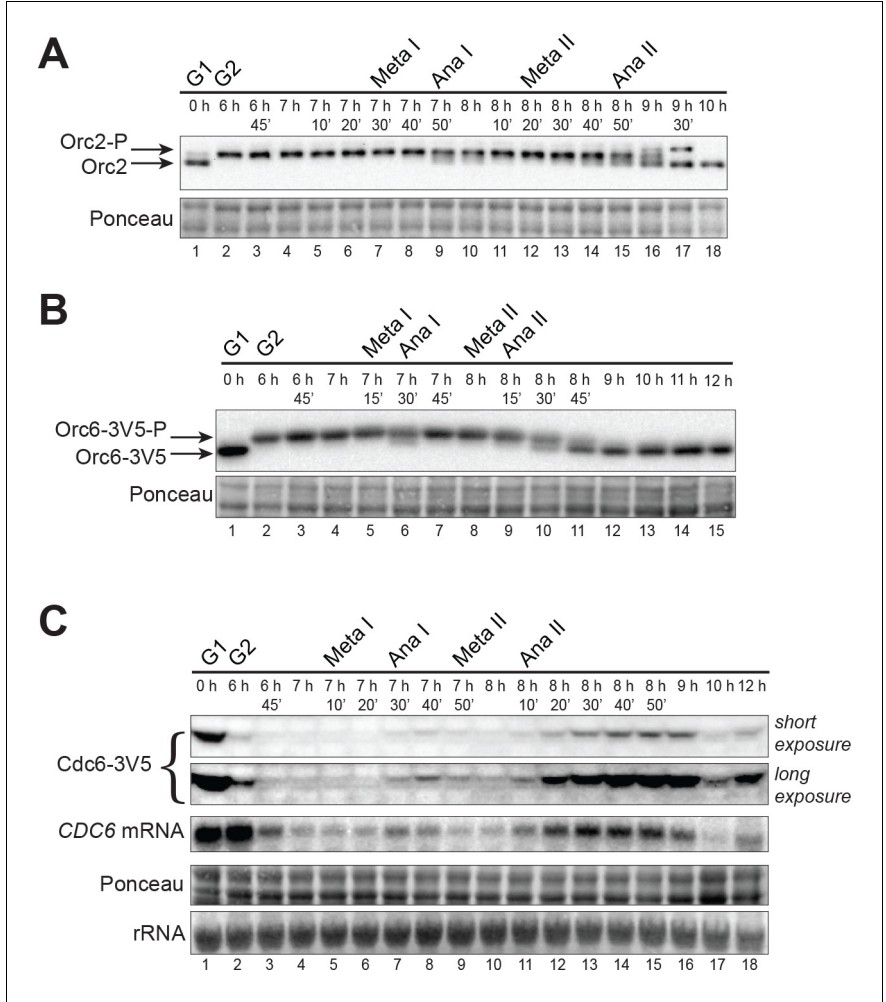

**Figure 2.** CDK-dependent inhibitory mechanisms are weakened during the MI–MII transition. (**A**) Orc2 (strain yDP71) and (**B**) Orc6 (strain yDP120) are both transiently dephosphorylated during the MI–MII transition. ORC was detected by immunoblot during meiosis. A phosphorylation–dependent shift in electrophoretic-mobility reveals Orc2 and Orc6 phosphorylation states. The time after transfer into sporulation medium and the associated meiotic stages are indicated above each lane. For cell–cycle stage quantification, see *Figure 1—figure supplement 1B* (Orc2) and *Figure 2—figure supplement 1A* (Orc6). (**C**) *CDC6* protein and mRNA transiently reaccumulate during the MI–MII transition (strain yDP71). Top: Cdc6 immunoblots (short and long exposures) during meiosis. Bottom: *CDC6* mRNA levels were detected by northern blots during meiosis. The time after transfer into sporulation medium and the associated meiotic stages are indicated above each lane. For cell–cycle stage quantification, see *Figure 2—figure supplement 1B*.

DOI: https://doi.org/10.7554/eLife.33309.007

The following figure supplement is available for figure 2:

**Figure supplement 1.** Cells from *Figure 2* proceeded synchronously through meiosis.

DOI: https://doi.org/10.7554/eLife.33309.008

*CDC6* transcription (*Moll et al., 1991*; *Piatti et al., 1995*). We found that Cdc6 protein was present during pre–meiotic G1 phase but became undetectable in MI when CDK is highly active (*Figure 2C*). During the MI–MII transition, however, Cdc6 protein partially reaccumulated (*Figure 2C* lane 8) before decreasing again in MII. As with Cdc6 protein levels, although *CDC6* mRNA was low during most of MI and MII, expression increased briefly during the MI–MII transition (*Figure 2C* lane 7). Taken together, these findings suggested that reduced CDK activity during the MI–MII transition results in a partial but detectable decrease of ORC phosphorylation and a slight reaccumulation of Cdc6.

# Ime2 inhibits the Mcm2–7 complex by an intrinsic mechanism to prevent helicase loading

The transient weakening of the CDK-dependent inhibitory mechanisms suggested the existence of meiosis–specific mechanisms to inhibit helicase loading. A strong candidate to mediate these potential mechanisms was Ime2, a CDK–related meiosis–specific kinase (*Krylov et al., 2003*). Previous studies found that Ime2 is active during the meiotic divisions (*Berchowitz et al., 2013*) and that, like CDK, this kinase can promote Mcm2–7 nuclear export upon completion of meiotic S phase (*Holt et al., 2007*). However, if Ime2 were replacing CDK–dependent inhibition of helicase loading during the MI–MII transition, we hypothesized that it would inhibit Mcm2–7 loading by more than one mechanism (as CDK is known to do).

To identify additional mechanisms by which Ime2 inhibits Mcm2–7 loading, we asked if Ime2 could inhibit helicase loading in vitro. To this end, we used an assay that reconstitutes helicase loading on origin–containing DNA with four purified proteins (ORC, Cdc6, Cdt1, and Mcm2–7; see reaction scheme in *Figure 3A*) (*Evrin et al., 2009*; *Remus et al., 2009*). We found that pre-treating the helicase-loading proteins with purified Ime2 fully inhibited Mcm2–7 loading (*Figure 3B and C*). To demonstrate that this inhibition depended on Ime2 kinase activity, we purified an analog–sensitive Ime2 protein (Ime2–AS). Analog–sensitive kinases are active in the presence of ATP but are inhibited by specific bulky ATP analogs (*Bishop et al., 2000*). In the case of Ime2–AS, addition of the ATP analog 1–NA–PP1 strongly inhibits its kinase activity (*Benjamin et al., 2003*). In the presence of ATP, Ime2–AS inhibited Mcm2–7 loading to the same extent as wild–type (WT) Ime2 (compare *Figure 3—figure supplement 1* to *Figure 3C*). However, the addition of 1–NA–PP1 to assays treated with Ime2–AS (but not WT Ime2) fully restored helicase loading (*Figure 3D* lanes 3–10). Consistent with Ime2 phosphorylation being responsible for the inhibition, the extent of helicase-loading inhibition correlated with the extent of phosphorylation of helicase–loading proteins for both WT Ime2 and Ime2–AS, with and without 1–NA–PP1 treatment (*Figure 3—figure supplement 2*). Taken together, these data demonstrate that Ime2-phosphorylation of one or more helicase-loading proteins directly inhibits origin licensing. Furthermore, because these experiments use only purified proteins, the mechanism preventing helicase loading must be due to the intrinsic inhibition of a specific protein's function, as opposed to indirect inhibitory mechanisms such as nuclear export or protein degradation.

To more precisely elucidate the mechanism of Ime2-inhibition of helicase loading, we sought to determine the step of this reaction that Ime2 inhibits. During helicase loading, origin–bound ORC–Cdc6 complexes recruit Cdt1–Mcm2–7 heptamers to form a short-lived complex called the OCCM (for ORC-Cdc6-Cdt1-Mcm2-7) (*Randell et al., 2006*; *Sun et al., 2013*; *Ticau et al., 2015*; *Yuan et al., 2017*). After OCCM formation, multiple conformational changes and ATP hydrolysis events are required for the first Mcm2–7 complex to be stably loaded around the DNA and a second Mcm2–7 to be recruited and loaded (*Coster and Diffley, 2017*; *Coster et al., 2014*; *Kang et al., 2014*; *Ticau et al., 2017*; *Yuan et al., 2017*; *Zhai et al., 2017*). To determine whether Ime2 inhibits initial Mcm2–7 recruitment, we conducted in vitro association assays with the slowly hydrolyzable analog ATPγS instead of ATP, stalling helicase loading after OCCM formation (reaction scheme in *Figure 3A*) (*Randell et al., 2006*). Prior phosphorylation of the helicase–loading proteins with Ime2 (in the presence of ATP) did not prevent subsequent OCCM formation in the presence of excess ATPγS (*Figure 3E*). Thus, Ime2 phosphorylation does not block the protein–protein interactions necessary for initial Mcm2–7 recruitment, but instead must inhibit a downstream step during Mcm2–7 loading.

Next, we sought to identify the Ime2-target(s) that result in the inhibition of helicase loading. In vitro kinase assays showed that Ime2 phosphorylates Cdc6, Cdt1, and subunits of both ORC and the Mcm2–7 complex (*Figure 4A*). To identify which of these phosphorylation events inhibits helicase loading, we used Ime2–AS to phosphorylate each of these proteins separately. We then inhibited Ime2–AS by the addition of 1–NA–PP1 before adding the three remaining, non–phosphorylated proteins and origin DNA to initiate helicase loading. Strikingly, we found that phosphorylation of the Mcm2–7 complex alone resulted in a >90% decrease in helicase loading (*Figure 4B and C*). In contrast, Ime2 phosphorylation of ORC resulted in a ~50% reduction in loading, whereas reactions with phosphorylated Cdc6 and Cdt1 showed only minor defects (*Figure 4B and C*). Although CDK also inhibits helicase loading in vitro, the equivalent experiment using CDK confirmed previous results

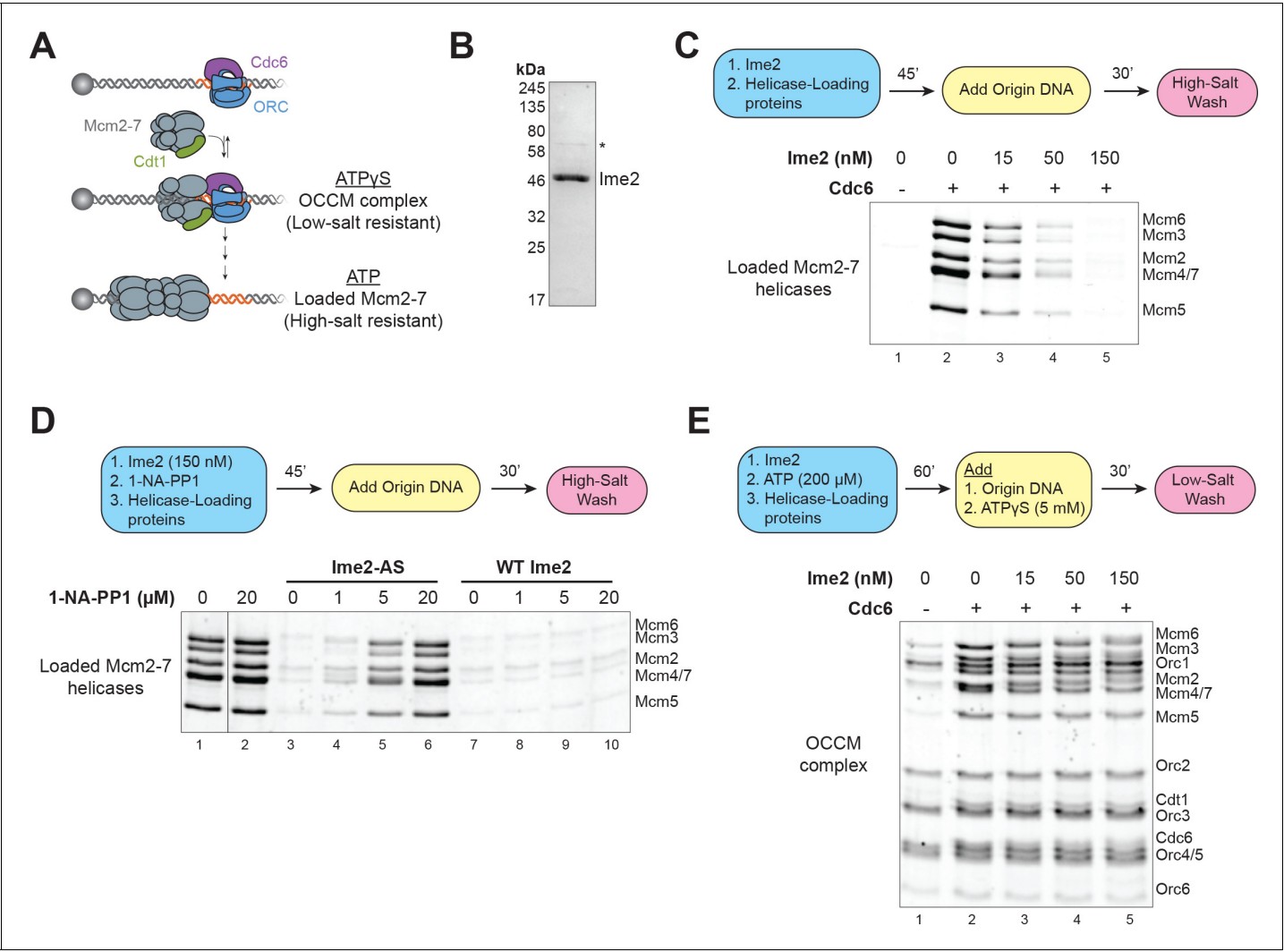

**Figure 3.** Ime2 is sufficient to inhibit helicase loading in vitro. (A) Diagram of helicase–loading and OCCM–complex–formation assays. Origin–containing DNA (red) is bound to a magnetic bead. Origin bound ORC–Cdc6 complexes recruit Cdt1–Mcm2–7 heptamers to form the OCCM complex. In ATPγS, the reaction stops at this point, and the whole complex is stable in low–salt washes. In ATP, helicase loading proceeds to completion resulting in Mcm2–7 complexes encircling the DNA that are stable in high–salt washes. (B) Purification of Ime2$^{stable}$–3XFlag. Asterisk (*) marks a slight contaminant. (C) Pre–incubation of Ime2 with the helicase–loading proteins inhibits Mcm2–7 loading onto replication origins in vitro. Top: Flowchart of experiment. Bottom: Helicase–loading assay at the indicated Ime2 concentration. Reaction lacking Cdc6 (lane 1) shows that Mcm2–7 complex DNA association depends on the helicase–loading reaction. (D) Ime2 inhibition of Mcm2–7 loading depends on its kinase activity. Top: Flowchart of experiment. Bottom: Helicase–loading assay. Purified Ime2–AS (150 nM) can inhibit Mcm2–7 loading (lane 3), and this inhibition can be prevented by increasing 1–NA–PP1 concentration (lanes 3–6). Wild–type Ime2 can inhibit Mcm2–7 loading regardless of 1–NA–PP1 concentration (lanes 7–10). (E) Ime2 cannot inhibit Mcm2–7-Cdt1 recruitment to ORC-Cdc6 in ATPγS. Top: Flowchart of experiment. Bottom: OCCM–complex–formation assay at the indicated Ime2 concentration (lanes 2–5). Mcm2–7-Cdt1 recruitment depends on Cdc6 (lane 1).

DOI: https://doi.org/10.7554/eLife.33309.009

The following figure supplements are available for figure 3:

**Figure supplement 1.** Ime2-AS inhibits helicase loading.

DOI: https://doi.org/10.7554/eLife.33309.010

**Figure supplement 2.** Ime2 kinase activity correlates with inhibition of helicase loading.

DOI: https://doi.org/10.7554/eLife.33309.011

showing that CDK only strongly inhibits ORC activity (*Figure 4—figure supplement 1*) (*Chen and Bell, 2011*). Together, these data demonstrate that Ime2 phosphorylation of the Mcm2–7 complex is sufficient to inhibit helicase loading. Furthermore, although Ime2 and CDK both directly inhibit helicase loading, the critical target required for their direct inhibition is distinct.

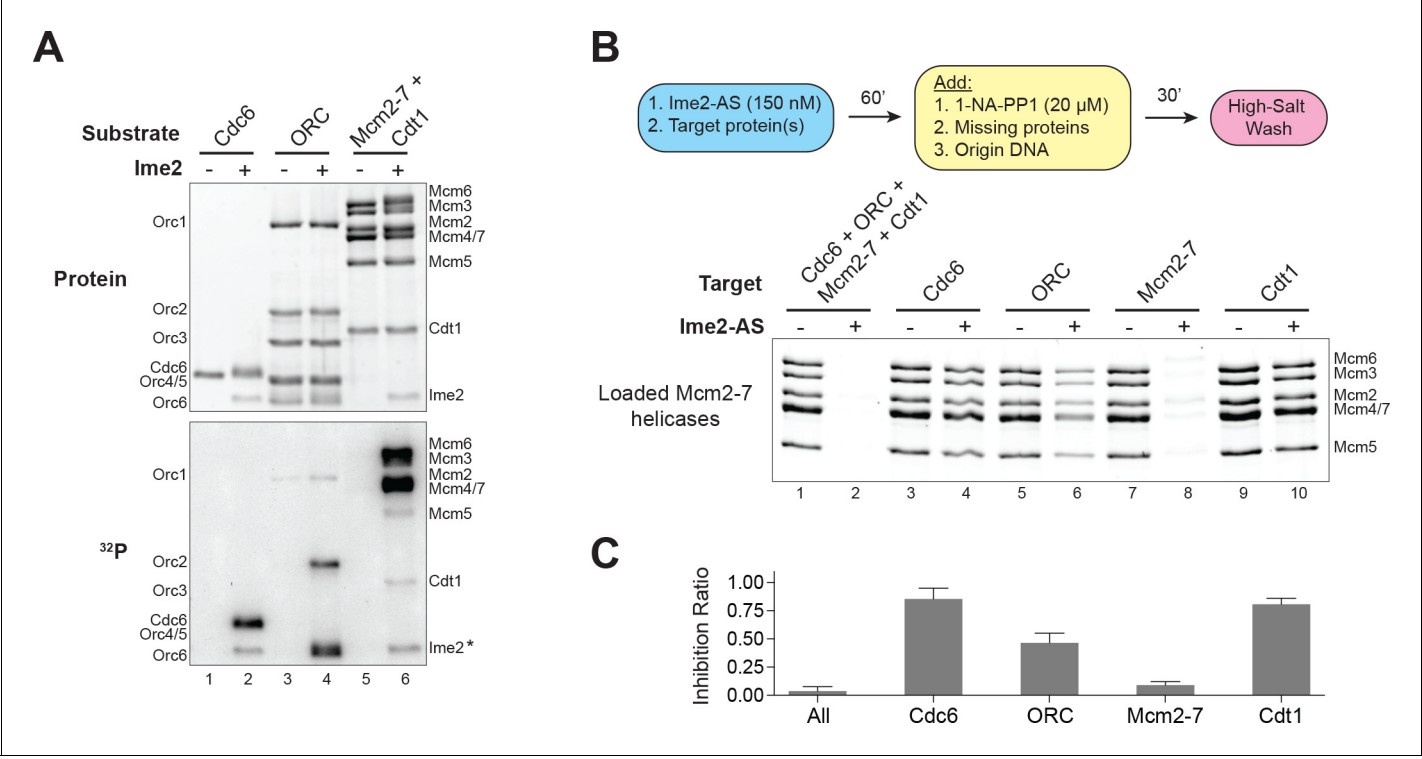

**Figure 4.** Ime2–phosphorylation of the Mcm2–7 complex intrinsically inhibits its loading onto replication origins. (**A**) Ime2 can phosphorylate Cdc6, ORC, Cdt1 and Mcm2–7 in vitro. Buffer control (lanes 1, 3, and 5) or 50 nM Ime2 (lanes 2, 4, and 6) were incubated with the indicated substrate proteins. The substrates were purified Cdc6 (lanes 1 and 2), ORC (lanes 3 and 4) or Mcm2–7–Cdt1 (lanes 5 and 6). Asterisk (*) marks Ime2 autophosphorylation. Top: total protein (Krypton stain). Bottom: phosphorylated protein (modified with [γ-$^{32}$P] ATP). (**B**) Ime2–phosphorylation of each protein separately shows that the primary target of Ime2–mediated inhibition is the Mcm2–7 complex (compare lanes 7 and 8). Top: Flowchart of experiment. Bottom: Helicase–loading assay after prior Ime2–phosphorylation of indicated protein. (**C**) Quantification of (**B**) from three independent experiments. Inhibition ratio was calculated as total Mcm2–7 loading from the +Ime2 reactions divided by amount of loading in the corresponding reaction lacking Ime2. The mean is represented by the height of the bar. Error bars represent the standard deviation of three independent experiments.

DOI: https://doi.org/10.7554/eLife.33309.012

The following source data and figure supplement are available for figure 4:

**Source data 1.** Raw values used for the quantification of *Figure 4C*.

DOI: https://doi.org/10.7554/eLife.33309.014

**Figure supplement 1.** CDK–phosphorylation of ORC intrinsically inhibits helicase loading.

DOI: https://doi.org/10.7554/eLife.33309.013

## CDK and Ime2 cooperate to inhibit Mcm2–7 loading and Cdc6 expression during the MI–MII transition

A critical question we sought to answer was which kinase inhibits helicase loading during the MI–MII transition in vivo. To address this question, we employed yeast strains with analog–sensitive alleles of CDK (*cdk1–as*) and Ime2 (*ime2–as*) in place of their respective wild-type alleles. As cells were entering the MI–MII transition (≥45% of cells in anaphase I; *Figure 5—figure supplement 1*), we inhibited these kinases and examined Mcm2–7 loading, *CDC6* mRNA and protein expression, and ORC phosphorylation. As a control, we confirmed that cells with wild–type *CDK1* and *IME2* were unaffected by addition of inhibitors (compare *Figure 5A* to *Figures 1* and *2*).

We inhibited CDK and Ime2 separately to determine the impact of each kinase on helicase loading. Inhibition of CDK at the end of MI promoted only limited Mcm2–7 reloading and reaccumulation of *CDC6* mRNA and protein, although ORC was substantially dephosphorylated in this condition (*Figure 5B*). In mitotic cells, CDK-inhibition bypasses all of these inhibitory mechanisms and promotes robust Mcm2–7 reloading (*Dahmann et al., 1995*; *Drury et al., 2000*; *Nguyen et al., 2001*), and we have recapitulated this result with the *cdk1–as* allele (*Figure 5—figure*

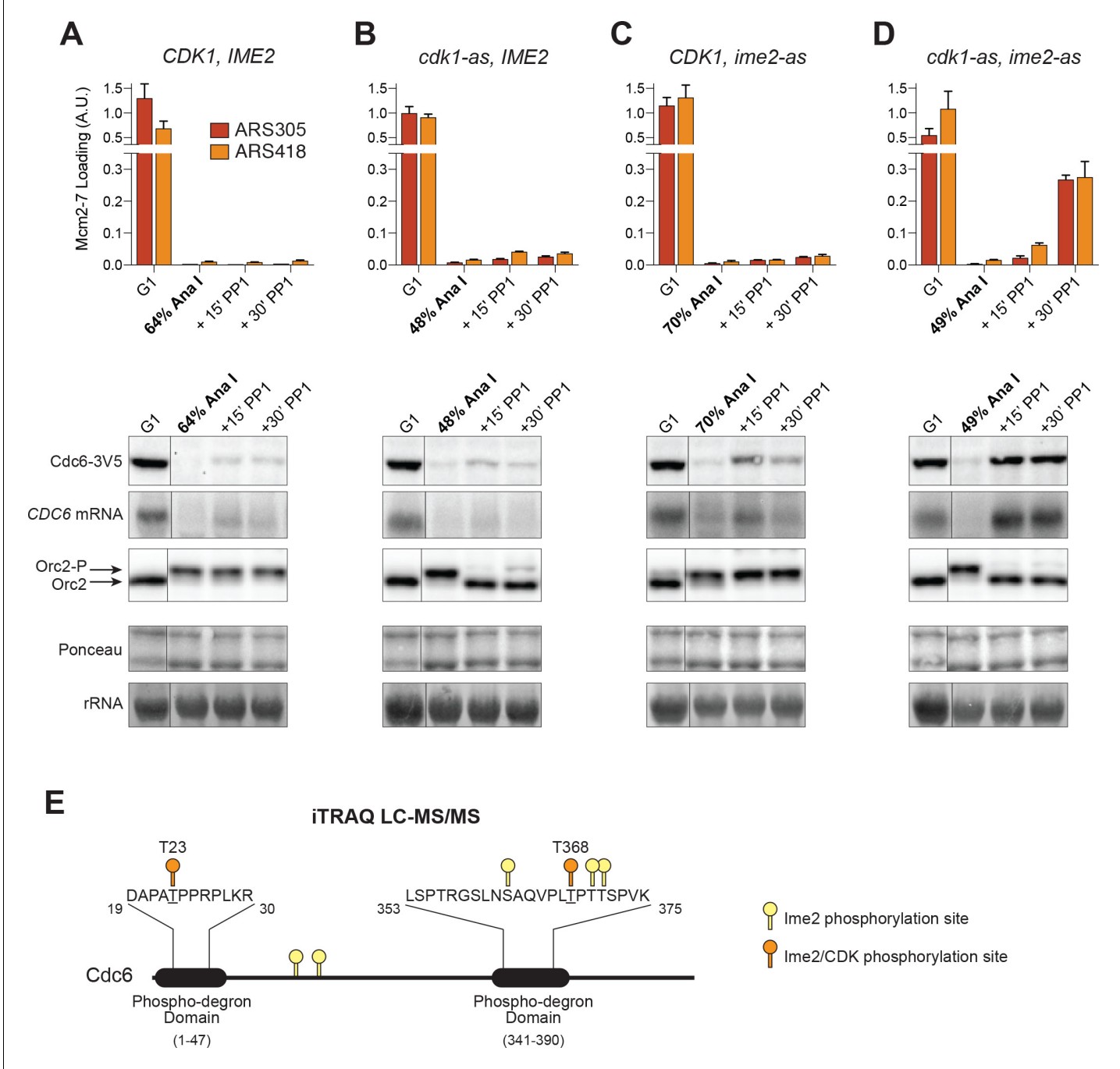

**Figure 5.** CDK and Ime2 cooperate to prevent Mcm2–7 loading and inhibit *CDC6* expression during the MI–MII transition. Simultaneous inhibition of both CDK and Ime2 is required for robust Mcm2–7 reloading and *CDC6* reaccumulation during the MI-MII transition. (A–D): Mcm2–7 loading (ChIP-qPCR), Orc2 phosphorylation (immunoblots), and *CDC6* protein and mRNA expression (immunoblots and northern blots) were analyzed in G1 phase as well as at the MI–MII transition. At the MI–MII transition, 10 μM 1–NM–PP1 and 20 μM 1–NA–PP1 were added. Samples were harvested 15 and 30 min after inhibitor addition. (A) Strain yDP71: *CDK1, IME2*. (B) Strain yDP152: *cdk1–as, IME2*. (C) Strain yDP176: *CDK1, ime2–as*. (D) Strain yDP177: *cdk1–as, ime2–as*. For cell–cycle stage quantification for *Figure 5A–5D*, see *Figure 5—figure supplement 1A–1D*, respectively. Mcm2–7 loading was analyzed at ARS305 (red) and ARS418 (orange). The peak % of input DNA immunoprecipitated (set to A.U. = 1.0) was 15.5% for ARS305 and 5.3% for ARS418. (E) Ime2 directly phosphorylates Cdc6 phospho–degron domains. Purified Cdc6 was treated with purified Ime2 or buffer–control in the presence of ATP. Quantitative mass spectroscopy was used to identify Ime2-dependent phosphorylation sites on Cdc6. Phosphorylation sites detected (with >4–fold enrichment upon Ime2 treatment) as well as the location of the Cdc6 phospho–degron domains are illustrated. Yellow markers indicate unique Ime2 sites. Orange markers indicate Ime2 sites that are also CDK sites based on previous work (*Calzada et al., 2000*; *Drury et al., 2000*). Phospho-degron domains are based on previous work (*Perkins et al., 2001*). For phosphorylation–site enrichment values, see *Supplementary file 1*.
*Figure 5 continued on next page*

*Figure 5 continued*

DOI: https://doi.org/10.7554/eLife.33309.015

The following source data and figure supplements are available for figure 5:

**Source data 1.** Raw values used for the quantification of *Figure 5A–5D*.

DOI: https://doi.org/10.7554/eLife.33309.019

**Figure supplement 1.** Cells in *Figure 5* entered the MI-MII transition at the time of kinase inhibition.

DOI: https://doi.org/10.7554/eLife.33309.016

**Figure supplement 2.** CDK inhibition using a *cdk1–as* allele is sufficient for Mcm2–7 reloading in mitotic cells.

DOI: https://doi.org/10.7554/eLife.33309.017

**Figure supplement 2—source data 1.** Raw values used for the quantification of *Figure 5—figure supplement 2*.

DOI: https://doi.org/10.7554/eLife.33309.018

*supplement 2*). Consequently, there must be CDK–independent mechanisms to inhibit origin licensing and *CDC6* expression that are specific to meiosis. To test whether Ime2 fulfills these functions, we inhibited Ime2 as cells were entering the MI–MII transition. Similar to CDK, however, Ime2-inhibition only caused limited Mcm2–7 reloading, although it did promote more significant reaccumulation of *CDC6* mRNA and protein than CDK-inhibition (*Figure 5C*). Therefore, neither CDK nor Ime2 are solely responsible for inhibiting helicase loading during the MI–MII transition.

Do CDK and Ime2 cooperate to repress helicase reloading? Strikingly, simultaneous inhibition of both CDK and Ime2 at the end of MI resulted in much higher levels of Mcm2–7 reloading than we observe with either kinase alone (*Figure 5D*). Furthermore, co–inhibition of these kinases restored expression of *CDC6* mRNA and protein to levels observed in pre–meiotic G1 cells (*Figure 5D*). Consistent with repression of Cdc6 by both kinases, mass spectrometry analysis showed that Ime2 phosphorylates multiple sites on Cdc6 within its phosphorylation–responsive degron domains in vitro, and two of these sites directly overlap with CDK–sites known to contribute to Cdc6 degradation (*Figure 5E*, *Supplementary file 1*) (*Calzada et al., 2000*; *Drury et al., 2000*). The transcriptional and proteolytic inhibition of Cdc6 by both CDK and Ime2 illustrates that it is a critical target of inhibition to prevent helicase loading. Thus, neither CDK nor Ime2 is capable of full inhibition of helicase loading but together they are a potent inhibitor of origin licensing and *CDC6* expression during the MI–MII transition.

## Cdc5 and CDK promote the degradation of Sld2, an essential helicase–activation protein

We considered the possibility that downstream steps of DNA replication could also be inhibited during the meiotic divisions. Despite the numerous mechanisms inhibiting Mcm2–7 loading, the weakening of ORC– and Cdc6–dependent controls (*Figure 2*) revealed a degree of leakiness in at least a subset of these mechanisms without any kinase perturbation. Therefore, we analyzed the abundance of proteins required for Mcm2–7 activation, which is the step after Mcm2–7 loading during replication initiation. Most helicase–activation proteins were present throughout the meiotic divisions, including Cdc45, Psf2 (a member of the GINS complex), Sld3, and Dpb11 (*Figure 6—figure supplement 1*). In contrast, Sld2 was robustly degraded upon entry into MI and did not reaccumulate until the completion of MII (*Figure 6A*, *Figure 6—figure supplement 2*). Sld2 is essential for replication initiation (*Kamimura et al., 1998*; *Yeeles et al., 2015*) and thus, its degradation represents a robust mechanism to inhibit helicase activation.

Both CDK and the polo–like kinase Cdc5 were candidates to regulate meiotic Sld2 protein levels (*Reusswig et al., 2016*). During mitotic divisions, total Sld2 protein levels do not change dramatically until the end of mitosis, at which point CDK– and Cdc5– phosphorylation sites within a phospho–degron domain on Sld2 become important for its degradation. In addition, Cdc5 and the M–phase cyclins Clb1, Clb3, and Clb4 are transcriptionally induced by the meiosis-specific transcription factor Ndt80 (*Chu and Herskowitz, 1998*), and both Cdc5 and CDK are active during the meiotic divisions (*Attner et al., 2013*; *Benjamin et al., 2003*; *Carlile and Amon, 2008*; *Clyne et al., 2003*; *Dahmann and Futcher, 1995*; *Lee and Amon, 2003*; *Sourirajan and Lichten, 2008*).

To test whether Cdc5 and CDK contribute to Sld2 degradation during meiosis, we mutated their previously identified phosphorylation sites on Sld2 (*Reusswig et al., 2016*). Mutation of either

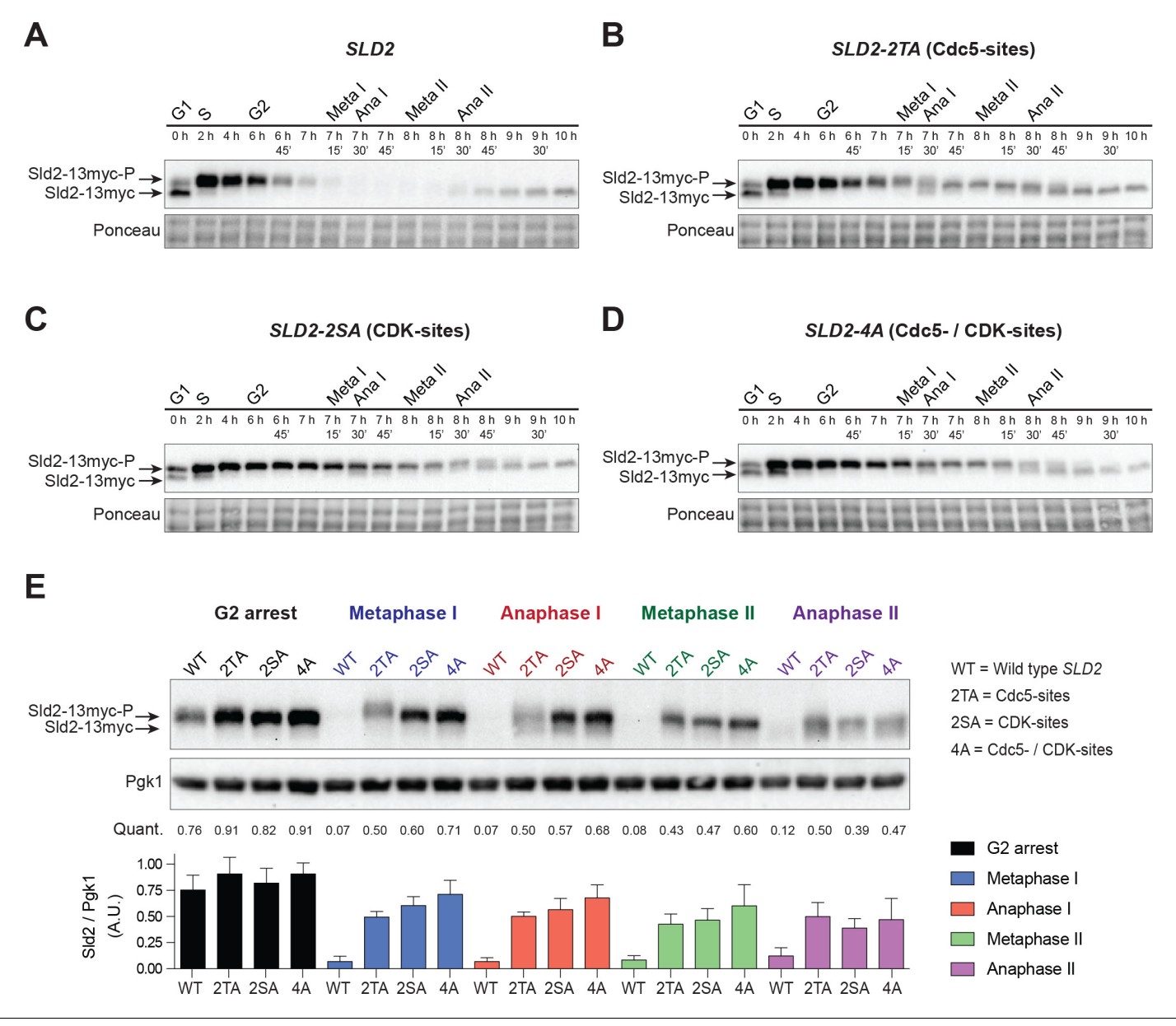

**Figure 6.** Sld2 is degraded during the meiotic divisions in a manner that depends on Cdc5- and CDK-phosphorylation sites. (A) Sld2 protein is degraded upon entry into the meiotic divisions. Immunoblots of Sld2–13myc during meiosis from strain yDP336. The time after transfer into sporulation medium and the associated meiotic stages are indicated above each lane. For cell–cycle synchrony, refer to *Figure 6—figure supplement 2A*. (B–D) Mutation of either Cdc5– or CDK–phosphorylation sites on Sld2 results in stabilization of Sld2 throughout the meiotic divisions: Immunoblots of Sld2– 13myc during meiosis with the following mutations: (B) Cdc5–phosphorylation sites (strain yDP473: 2TA – T122A/T143A), (C) CDK–phosphorylation sites (strain yDP642: 2SA – S128A/S138A), or (D) Cdc5– and CDK–phosphorylation sites (strain yDP644: 4A – T123A/S128A/S138A/T143A). The time after transfer into sporulation medium and the associated meiotic stages are indicated above each lane. For cell–cycle synchrony, refer to *Figure 6—figure supplement 2B–2D*. (E) Top: Samples from (A–D) with the peak number of cells in G2, Metaphase I, Anaphase I, Metaphase II, and Anaphase II were run side–by–side. Middle: Mean of Sld2 levels normalized to PGK1 levels from three independent experiments. Bottom: Graph of Sld2/PGK1 quantification from three independent experiments. The mean is represented by the height of the bar. Error bars represent the standard deviation.
DOI: https://doi.org/10.7554/eLife.33309.020

The following source data and figure supplements are available for figure 6:

**Source data 1.** Raw values used for the quantification of *Figure 6E*.
DOI: https://doi.org/10.7554/eLife.33309.025

**Figure supplement 1.** Most helicase–activation proteins are present throughout the meiotic divisions.
DOI: https://doi.org/10.7554/eLife.33309.021

*Figure 6 continued on next page*

*Figure 6 continued*

**Figure supplement 2.** Cells from *Figure 6* proceeded synchronously through meiosis.
DOI: https://doi.org/10.7554/eLife.33309.022
**Figure supplement 3.** Sld2-13myc and Pgk1 levels are quantifiable across a 32-fold dilution range.
DOI: https://doi.org/10.7554/eLife.33309.023
**Figure supplement 3—source data 1.** Raw values used for the quantification of *Figure 6—figure supplement 3*.
DOI: https://doi.org/10.7554/eLife.33309.024

Cdc5–phosphorylation sites (*sld2-2TA* = T122A and T143A), CDK–phosphorylation sites (*sld2-2SA* = S128A and S138A), or both together (*sld2-4A*) resulted in stabilization of Sld2 during the meiotic divisions (*Figure 6B–D*, *Figure 6—figure supplement 2*). To directly compare the effects of these mutations, we ran samples from all four strains side–by–side at each stage of meiosis and quantified Sld2 levels relative to a PGK1 loading control (*Figure 6E*, *Figure 6—figure supplement 3*). No more than a 2–fold difference in Sld2 levels was detected during meiotic G2 between all four strains. From metaphase I until metaphase II, wild type (WT) Sld2 abundance decreased to ~10% of the levels observed in G2 phase. In contrast, the abundance of the Sld2-4A mutant remained almost unchanged during this time. The levels of the Sld2-2TA and Sld2-2SA protein were marginally less than the Sld2-4A protein at these same times. Because both Sld2-2SA or Sld2-2TA were expressed at much higher levels than the WT protein during the meiotic divisions, both CDK– and Cdc5–phosphorylation are required to drive substantial Sld2 degradation. Upon entry into anaphase II, WT Sld2 protein levels began to recover while the mutant protein levels decreased, suggesting that other mechanisms were impacting Sld2 expression. Together, these data indicate that CDK and Cdc5 inhibit Mcm2–7 activation during the meiotic divisions by promoting Sld2 degradation. Thus, if an origin escapes the inhibition of Mcm2-7 loading during the meiotic divisions, Sld2 degradation would prevent activation of the associated helicases.

## Discussion

In this study, we address a fundamental question concerning the regulation of meiosis; how do meiotic cells undergo two sequential rounds of chromosome segregation without an intervening S–phase? We found that meiotic cells prevent DNA replication between the meiotic divisions using CDK, Ime2, and Cdc5 to inhibit both helicase loading and activation. Ime2 and CDK cooperate to inhibit helicase loading, and their co–inhibition was sufficient for aberrant origin licensing during the MI–MII transition. Compared with CDK, Ime2 inhibits origin licensing using both overlapping and distinct mechanisms. In particular, we found that unlike CDK, Ime2 phosphorylation of Mcm2–7 directly inhibits its participation in helicase loading. In addition to the inhibition of origin licensing, meiotic cells use CDK and the polo–like kinase Cdc5 to promote degradation of Sld2, a key helicase-activation protein. Together, these data reveal that multiple kinases inhibit DNA replication between the two meiotic divisions by targeting both many components of the helicase-loading machinery and at least one helicase-activation protein. These mechanisms combine to ensure the hallmark reduction in ploidy associated with meiotic cell division.

Elements of both the CDK–balance model (*Iwabuchi et al., 2000*) and the alternative–kinase model (*Holt et al., 2007*) are evident in our results (*Figure 7A*). In support of the CDK–balance model, the amount of CDK activity present during the MI–MII transition is sufficient to inhibit most origin licensing, because Ime2 inhibition does not result in strong Mcm2–7 reloading. Thus, the decrease in CDK activity that resets the chromosome segregation program (*Bizzari and Marston, 2011*; *Buonomo et al., 2003*; *Carlile and Amon, 2008*; *Fox et al., 2017*; *Marston et al., 2003*) is not sufficient to fully reset the DNA replication program. In support of the alternative–kinase model, we found that CDK inhibition similarly did not result in extensive Mcm2–7 reloading during the MI-MII transition. Instead, we show that two additional kinases contribute to the inhibition of DNA replication between the meiotic divisions. Ime2 inhibits helicase loading, and Cdc5 helps stimulate the degradation of Sld2.

The inhibition of DNA replication by Ime2 and Cdc5 allows for CDK to oscillate and reset the chromosome segregation program without allowing a new round of DNA replication. That meiotic cells use Ime2 and Cdc5 to inhibit helicase loading and activation, respectively, suggests that CDK–

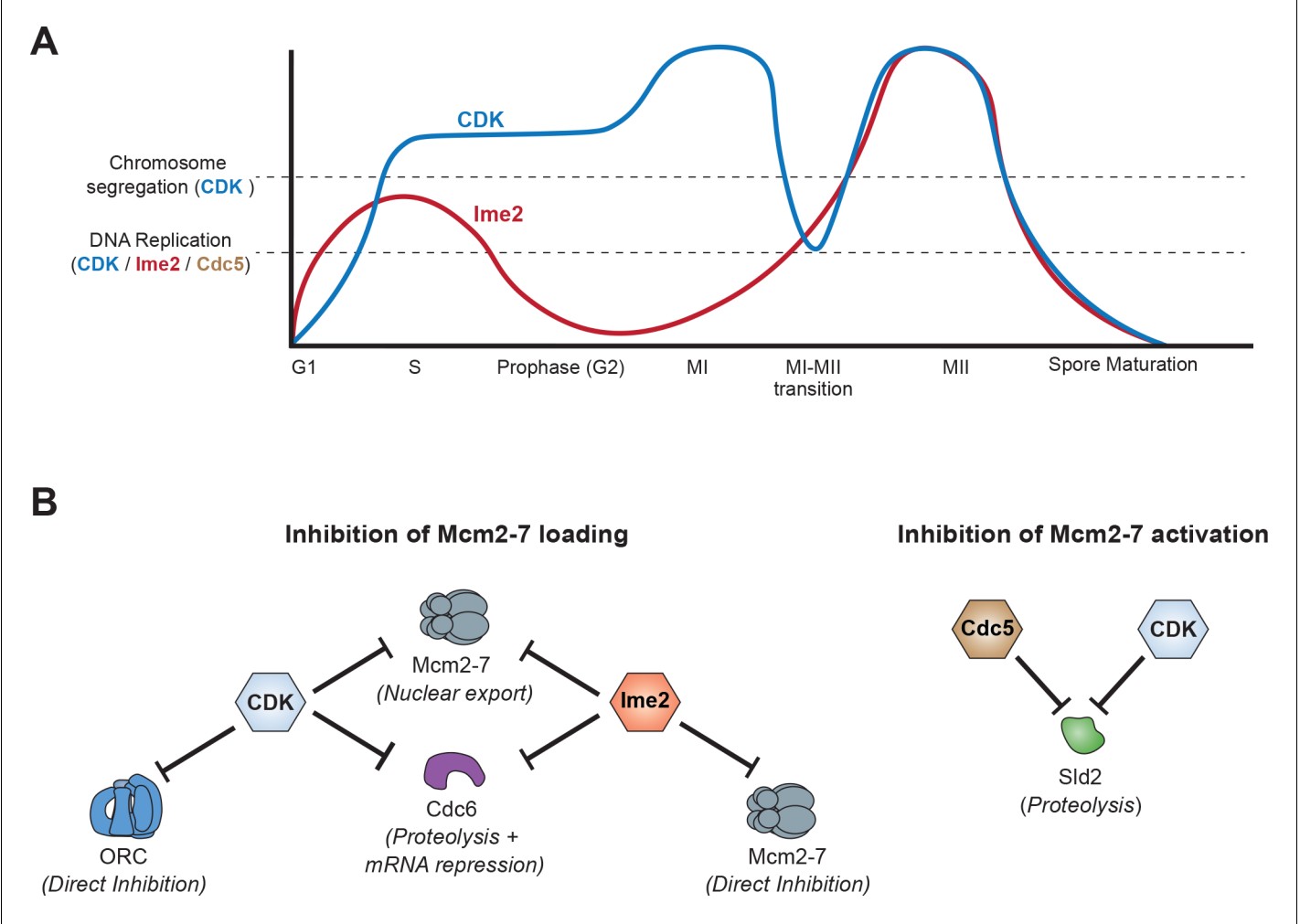

**Figure 7.** Model of how meiotic cells inhibit DNA replication during the MI–MII transition. (**A**) Graphical representation of CDK (blue) (*Carlile and Amon, 2008*) and Ime2 (red) (*Berchowitz et al., 2013*) kinase activities during meiosis, and how they regulate the chromosome segregation and DNA replication programs. Chromosome segregation is regulated by CDK, whereas DNA replication is regulated by CDK, Ime2, and Cdc5. During the MI-MII transition, CDK activity decreases enough to reset the chromosome segregation program for MII. Although CDK remains active enough to mostly inhibit the DNA replication program, the decreased activity is a significant threat to the inhibition of origin licensing. Ime2 is also mostly sufficient to inhibit Mcm2-7 loading during the MI-MII transition. Cdc5 activity has not been precisely determined and is thus not shown, but it contributes to the inhibition of DNA replication by limiting Mcm2–7 activation. (**B**) The mechanisms and effector proteins used to inhibit DNA replication during the meiotic divisions. CDK and Ime2 cooperate to inhibit helicase loading by promoting Mcm2–7 nuclear export and the repression of *CDC6* by proteolytic degradation and transcriptional inhibition. Additionally, Ime2 phosphorylates and directly inhibits the Mcm2–7 complex, whereas CDK directly inhibits ORC. To inhibit helicase activation, CDK and Cdc5 promote the proteolytic degradation of Sld2.
DOI: https://doi.org/10.7554/eLife.33309.026

dependent inhibition of origin licensing is not adequate to ensure genome stability during meiosis. Furthermore, a subset of the mechanisms by which CDK and Ime2 inhibit helicase loading are non–overlapping, providing additional avenues to prevent inappropriate replication (*Figure 4*). It is important to note that our helicase reloading results are population–based experiments. Accordingly, the slight helicase reloading and Cdc6 reaccumulation that occurred upon inhibition of each kinase separately suggests that, at some point during the MI-MII transition or at a subset of replication origins, each kinase is required to prevent origin licensing.

Why do meiotic cells use so many mechanisms to inhibit DNA replication between the meiotic divisions (*Figure 7B*)? We speculate that this is due to the difficulty of preventing >300 replication origins from initiating while CDK activity oscillates. Emphasizing the importance of completely inhibiting DNA replication at unwanted periods of the cell cycle, previous studies have shown that

reinitiating replication from even a single origin can cause gene amplification and chromosome mis-segregation (*Green et al., 2010*; *Hanlon and Li, 2015*). The transient weakening of both Cdc6 repression and ORC phosphorylation during the MI–MII transition (*Figure 2*) indicates that the associated decrease in CDK activity (*Carlile and Amon, 2008*) is a significant threat to origin licensing. Even the additional repression of Cdc6 by Ime2 is not sufficient to fully eliminate the protein during the MI-MII transition, although Ime2 has a larger role than CDK for Cdc6 repression at this time (*Figure 5*). It is worth noting that in multicellular eukaryotes, preventing DNA re-replication in meiotic cells is more important than in mitotic cells to ensure that the inherited genome remains intact.

In addition to helicase loading, our studies strongly suggest that meiotic cells inhibit downstream steps of DNA replication. In mitotic cells, prevention of DNA re–replication relies on inhibiting Mcm2–7 loading (*Arias and Walter, 2007*). Inhibiting early steps of replication makes sense, as there is less danger of aberrant DNA unwinding and polymerase recruitment. Although helicase loading is an earlier event in replication initiation, we note that Sld2 is required to form the active eukaryotic replicative helicase, the Cdc45–Mcm2–7–GINS (CMG) complex (*Heller et al., 2011*; *Kamimura et al., 1998*; *Muramatsu et al., 2010*; *Yeeles et al., 2015*). Activation of the replicative helicase is the committed step of replication initiation (*Bell and Labib, 2016*) and preventing this step would still stop replication before initial DNA unwinding and synthesis. Previous studies found that Sld2 degradation in mitotic cells is important for ensuring genome stability at the M–G1 transition, but not during other parts of the mitotic cell cycle (*Reusswig et al., 2016*). Our finding that Sld2 is absent during both meiotic divisions suggests that mechanisms preventing DNA re–replication at the mitotic M–G1 transition also function during meiosis at the MI–MII transition, a partially G1-like state. Consistent with this idea, Dbf4, which binds to Cdc7 to form the kinase DDK and is also required for helicase activation, is degraded during both mitotic anaphase (*Ferreira et al., 2000*) as well as meiotic anaphase I (*Matos et al., 2008*). The degradation of these two proteins strongly suggests that meiotic cells further protect themselves from replication initiation during the meiotic divisions by inhibiting multiple steps during Mcm2–7 activation.

Ime2 is not just a backup for the known inhibitory mechanisms used by CDK. We found that in addition to using at least two mechanisms similar to CDK (Cdc6 degradation (*Figure 5*) (*Drury et al., 2000*) and Mcm2-7 nuclear export (*Holt et al., 2007*)), Ime2-phosphorylation of Mcm2-7 results in a distinct mechanism of inhibition not seen after CDK-phosphorylation (*Figure 4*). Origin licensing is inhibited by a number of different mechanisms across eukaryotic evolution, but previous studies have not identified a mechanism that directly prevents Mcm2-7 from completing helicase loading (*Arias and Walter, 2007*; *Siddiqui et al., 2013*). Understanding how helicase loading is inhibited by CDK-phosphorylation of ORC and Ime2-phosphoryation of Mcm2-7 will reveal whether these two kinases target the same or different steps of this process.

The ability of Ime2 to inhibit helicase loading also suggests an important role for this kinase during the meiotic G1–S transition. During the mitotic G1–S transition, helicase loading is inhibited by G1–CDK before helicase activation is stimulated by S–phase CDK (*Drury et al., 2000*; *Labib et al., 1999*). This 'insulation' prevents cells from being in a state that permits both helicase loading and activation (e.g. at an intermediate level of CDK activity). During meiosis, G1 cyclins are not expressed and it is Ime2 that triggers activation of S–CDK at the G1–S transition (*Benjamin et al., 2003*; *Dirick et al., 1998*). Our data and previous studies (*Holt et al., 2007*) show that Ime2 can robustly inhibit helicase loading by multiple mechanisms, similar to G1-CDK. These observations strongly suggest that Ime2 insulates helicase loading from helicase activation during the meiotic G1–S transition in an analogous manner to G1–CDK during the mitotic G1–S transition.

The problem of uncoupling DNA replication and chromosome segregation during meiosis is conserved in other eukaryotes. Metazoans also use CDK to regulate both chromosome segregation and DNA replication (*Arias and Walter, 2007*; *Malumbres and Barbacid, 2009*; *Siddiqui et al., 2013*). Although metazoans mostly rely on CDK–independent mechanisms to inhibit Mcm2–7 loading during S and G2 phase, these mechanisms are not as potent during M–phase at which point Cdk1 becomes a critical inhibitor of origin licensing (*Arias and Walter, 2007*; *Siddiqui et al., 2013*). Transient inactivation of Cdk1 in human cells is sufficient for DNA re–replication, just as it is in yeast (*Bates et al., 1998*; *Dahmann et al., 1995*; *Itzhaki et al., 1997*). An oscillation of Cdk1 activity is also required for correct chromosome segregation in mammalian cells (*Malumbres and Barbacid, 2009*). With regard to how DNA replication and chromosome segregation are uncoupled during mammalian meiosis, previous studies raise the intriguing possibility that mammalian Cdk2 may have

a similar role during meiosis as Ime2 does in yeast. Human Cdk2 can rescue some meiotic defects associated with ime2Δ in yeast (*Szwarcwort-Cohen et al., 2009*). Additionally, Cdk2 is not required for mitosis in mice, but it is required for both male and female meiosis (*Ortega et al., 2003*). Together, our results provide new insight into how meiotic cells use multiple cell–cycle regulators and mechanisms to uncouple DNA replication and chromosome segregation.

# Materials and methods

## Key resources table

| Reagent type (species) or resource | Designation | Source or reference | Identifiers | Additional information |
|---|---|---|---|---|
| strain, strain background (*Saccharomyces cerevisiae* SK1) | yDP71 | This paper | Cdc6-3V5 | *SK1 MATa/alpha ura3::pGPD1-GAL4(848).ER::URA3/ura3::pGPD1- GAL4(848).ER::URA3 GAL-NDT80::TRP1/GAL-NDT80::TRP1 CDC6-3V5::KANMX6/CDC6-3V5::KANMX6* |
| strain, strain background (*S. cerevisiae* SK1) | yDP120 | This paper | Orc6-3V5 | *SK1 MATa/alpha ura3::pGPD1-GAL4(848).ER::URA3/ura3::pGPD1- GAL4(848).ER::URA3 GAL-NDT80::TRP1/GAL-NDT80::TRP1 ORC6-3V5::KANMX6/ORC6-3V5::KANMX6* |
| strain, strain background (*S. cerevisiae* SK1) | yDP152 | This paper | cdk1-as, Cdc6-3V5 | *SK1 MATa/alpha ura3::pGPD1-GAL4(848).ER::URA3/ura3::pGPD1- GAL4(848).ER::URA3 GAL-NDT80::TRP1/GAL-NDT80::TRP1 cdc28-as1(F88G)/cdc28-as1(F88G) CDC6-3V5::KANMX6/CDC6-3V5::KANMX6* |
| strain, strain background (*S. cerevisiae* W303) | yDP159 | This paper | Ime2 Purification | *W303 MATa bar1::hisG pep4::unmarked LEU2::pGAL1,10-IME2(1–404)–3xFLAG* |
| strain, strain background (*S. cerevisiae* SK1) | yDP176 | This paper | ime2-as, Cdc6-3V5 | *SK1 MATa/alpha ura3::pGPD1-GAL4(848).ER::URA3/ura3::pGPD1- GAL4(848).ER::URA3 GAL-NDT80::TRP1/GAL-NDT80::TRP1 ime2-as1(M146G)/ime2-as1(M146G) CDC6-3V5::KANMX6/CDC6-3V5::KANMX6* |
| strain, strain background (*S. cerevisiae* SK1) | yDP177 | This paper | cdk1-as, ime2-as, Cdc6-3V5 | *SK1 MATa/alpha ura3::pGPD1-GAL4(848).ER::URA3/ura3::pGPD1- GAL4(848).ER::URA3 GAL-NDT80::TRP1/GAL-NDT80::TRP1 cdc28-as1(F88G)/cdc28-as1(F88G) ime2-as1(M146G)/ime2-as1(M146G) CDC6-3V5::KANMX6/CDC6-3V5::KANMX6* |
| strain, strain background (*S. cerevisiae* SK1) | yDP329 | This paper | Dpb11-3V5 | *SK1 MATa/alpha ura3::pGPD1-GAL4(848).ER::URA3/ura3::pGPD1- GAL4(848).ER::URA3 GAL-NDT80::TRP1/GAL-NDT80::TRP1 DPB11-3V5::KANMX6/DPB11-3V5::KANMX6* |
| strain, strain background (*S. cerevisiae* SK1) | yDP330 | This paper | Psf2-3V5 | *SK1 MATa/alpha ura3::pGPD1-GAL4(848).ER::URA3/ura3::pGPD1- GAL4(848).ER::URA3 GAL-NDT80::TRP1/GAL-NDT80::TRP1 PSF2-3V5::KANMX6/PSF2-3V5::KANMX6* |
| strain, strain background (*S. cerevisiae* SK1) | yDP335 | This paper | Cdc45-13myc | *SK1 MATa/alpha ura3::pGPD1-GAL4(848).ER::URA3/ura3::pGPD1- GAL4(848).ER::URA3 GAL-NDT80::TRP1/GAL-NDT80::TRP1 CDC45-13myc::KANMX6/CDC45-13myc::KANMX6* |
| strain, strain background (*S. cerevisiae* SK1) | yDP336 | This paper | Sld2-13myc | *SK1 MATa/alpha ura3::pGPD1-GAL4(848).ER::URA3/ura3::pGPD1- GAL4(848).ER::URA3 GAL-NDT80::TRP1/GAL-NDT80::TRP1 SLD2-13myc::KANMX6/SLD2-13myc::KANMX6* |
| strain, strain background (*S. cerevisiae* SK1) | yDP337 | This paper | Sld3-13myc | *SK1 MATa/alpha ura3::pGPD1-GAL4(848).ER::URA3/ura3::pGPD1- GAL4(848).ER::URA3 GAL-NDT80::TRP1/GAL-NDT80::TRP1 SLD3-13myc::KANMX6/SLD3-13myc::KANMX6* |
| strain, strain background (*S. cerevisiae* SK1) | yDP473 | This paper | Sld2-2TA-13myc | *SK1 MATa/alpha ura3::pGPD1-GAL4(848).ER::URA3/ura3::pGPD1- GAL4(848).ER::URA3 GAL-NDT80::TRP1/GAL-NDT80::TRP1 SLD2(T122A T143A)–13myc::KANMX6/ SLD2(T122A T143A)–13myc::KANMX6* |

*Continued on next page*

*Continued*

| Reagent type (species) or resource | Designation | Source or reference | Identifiers | Additional information |
|---|---|---|---|---|
| strain, strain background (*S. cerevisiae* W303) | yDP554 | This paper | Ime2-AS Purification | *W303 MATa bar1::hisG pep4::unmarked LEU2::pGAL1,10-IME2(1–404, M146G)–3xFLAG* |
| strain, strain background (*S. cerevisiae* SK1) | yDP642 | This paper | Sld2-2SA-13myc | *SK1 MATa/alpha ura3::pGPD1-GAL4(848). ER::URA3/ura3::pGPD1- GAL4(848).ER::URA3 GAL-NDT80::TRP1/GAL-NDT80::TRP1 SLD2(S128A S138A)–13myc::KANMX6/ SLD2(S128A S138A)–13myc::KANMX6* |
| strain, strain background (*S. cerevisiae* SK1) | yDP644 | This paper | Sld2-4A-13myc | *SK1 MATa/alpha ura3::pGPD1-GAL4(848). ER::URA3/ura3::pGPD1- GAL4(848).ER::URA3 GAL-NDT80::TRP1/GAL-NDT80::TRP1 SLD2(T122A S128A S138A T143A)–13myc:: KANMX6/SLD2(T122A S128A S138A T143A) –13myc::KANMX6* |
| strain, strain background (*S. cerevisiae* W303) | ySK119 | Lõoke *et al*, 2017 (PMID: 28270517) | Cdk1-Clb5 Purification | W303 MATa bar1::hisG pep4::unmarked URA3::pGAL1,10-Cdc28-His,Δ1–95-Clb5-Flag |
| strain, strain background (*S. cerevisiae* SK1) | yST135 | This paper | Mcm2-7 Purification | *W303 MATa bar1::hisG pep4::unmarked TRP1::pSKM003(pGAL1,10-MCM6,MCM7) HIS3::pSKM004-(pGAL1,10-MCM2,Flag-MCM3) LYS2::pSKM002-(pGAL1,10-MCM4,MCM5)* |
| strain, strain background (*S. cerevisiae* W303) | yST144 | *Ticau et al. (2015)* (PMID: 25892223) | Mcm2-7-Cdt1 Purification | *W303 MATa bar1::hisG pep4::unmarked TRP1::pSKM003(pGAL1,10-MCM6,MCM7) HIS3::pSKM004-(pGAL1,10-MCM2,Flag-MCM3) LYS2::pSKM002-(pGAL1,10-MCM4,MCM5) URA3::pALS1(pGAL1,10-Cdt1,GAL4)* |
| strain, strain background (*S. cerevisiae* W303) | A4370 | Angelika Amon | cdk1-as | *W303 MATa bar1::hisG cdc28-as1(F88G)* |
| strain, strain background (*S. cerevisiae* W303) | ySDORC | John Diffley | ORC purification | *W303 MATa bar1::hyg pep4::kanMX TRP1::pGAL1,10-ORC5,ORC6 HIS3::pGAL1,10-ORC3,ORC4 URA3::pGAL1, 10-CBP-TEV-ORC1,ORC2* |
| antibody | poly ORC (Orc1 and Orc2 western blots and ORC ChIP) | | HM1108 (Bell Lab) | |
| antibody | Cdt1 | | HM5353 (Bell Lab) | |
| antibody | poly MCM (Mcm3 and Mcm6 western blots) | | UM174 (Bell Lab) | |
| antibody | poly MCM (Mcm2-7 ChIP) | | UM185 (Bell Lab) | |
| antibody | Mcm2 | | Santa Cruz, yN-19 (code sc-6680) | RRID:AB_648843 |
| antibody | Mcm7 | | Santa Cruz, yN-19 (code SC-6688) | RRID:AB_647936 |
| antibody | PGK1 | | Invitrogen (catalog #459250) | RRID:AB_2532235 |
| recombinant DNA reagent (plasmid) | pSKM033 | *Kang et al. (2014)* (PMID: 25087876) | Cdc6 purification | pGEX-GST-3C-FLAG-*CDC6* |
| recombinant DNA reagent (plasmid) | pALS16 | This study | Cdt1 purification | pGEX-GST-3C-*CDT1* |
| chemical compound, drug | 1-NM-PP1 | | Toronto Research Chemicals (catalog #A603003) | |
| chemical compound, drug | 1-NA-PP1 | | Cayman Chemical Co., (catalog #NC1049860) | |

## Yeast strains and plasmids

All *S. cerevisiae* strains are summarized in the Key Resources Table. Strains used for meiosis were diploids isogenic with SK1 *ho::LYS2/ho::LYS2, lys2/lys2, ura3/ura3, leu2::hisG/leu2::hisG, his3:hisG/his3:hisG, trp1::hisG/trp1::hisG*. Other strains were isogenic with W303 *ade2–1 trp1–1 leu2–3112 his3–11,15 ura3–1 can1–100*. Epitope tagging was done by homologous recombination as previously described (*Longtine et al., 1998*). Protein expression plasmids are described in the Key Resources Table.

## Meiotic time–courses

Meiotic time courses were done as described (*Berchowitz et al., 2013*). Briefly, saturated YPD cultures were diluted to an $OD_{600}$ = 0.25 in BYTA medium and grown for 18–22 hr. Cells were then washed once with water and resuspended to $OD_{600}$ = 1.9 in Sporulation medium before taking the 0–hour sample. All strains used for meiotic time courses included $P_{GPD}$–GAL4–ER, $P_{GAL}$–NDT80. Cultures were shaken at 30°C for 6 hr to allow cells to accumulate at the *NDT80* block. Addition of 1 μM β–estradiol (5 mM stock in ethanol [Sigma, E2758]) released cells from the *NDT80* block. Protein, RNA, ChIP, and immunofluorescence samples were harvested in parallel at the indicated time points.

## Immunofluorescence

Tubulin immunofluorescence was performed as described (*Berchowitz et al., 2013*). For spindle and nuclei scoring, 100 cells were counted per time point. Metaphase–I cells were defined as having a short, thick, bipolar spindle; Anaphase–I cells as having a long, bipolar spindle; Metaphase–II cells as having two short, thick, bipolar spindles; and Anaphase–II cells as having two long, bipolar spindles. Nuclei were counted as the number of separated DNA masses.

## Immunoblots

Cells were pelleted at 136,000 x g, resuspended in 5% TCA, and left at 4°C overnight. Pellets were washed with acetone, dried, and resuspended in 100 μL of 50 mM Tris [pH = 7.6],1 mM EDTA, 2.75 mM DTT, 1 mM PMSF, and 1x cOmplete Protease Inhibitors (Roche). Cells were lysed three times with glass beads using a FastPrep (MP Biomedicals), and boiled for 5 min after addition of 75 uL 5x sample buffer. Ponceau staining was used as a loading control. Proteins were detected with antibodies recognizing the epitope indicated or with the following antibodies: Orc1 and Orc2 (HM1108), Cdt1 (HM5353), Mcm3 and Mcm6 (UM174), Mcm2 (Santa Cruz, yN–19), Mcm7 (Santa Cruz, yN–19), and PGK1 (Invitrogen).

## Northern blots

Total RNA was isolated using a (400 μL:400 μL) mixture of TES buffer (10 mM Tris [pH7.6], 10 mM EDTA, 0.5% SDS) and acid phenol while shaking (Thermomixer, Eppendorf) with glass beads at 65°C for 30 min. After ethanol precipitation and resuspension in DEPC–treated water, equal amounts of RNA (between 10–14 μg) were loaded in each lane. rRNA was used as a loading control and detected with methylene blue. *CDC6*–specific [32]P–labeled probes were made by Klenow extension (GE Healthcare, RPN1605) Template: 3′–end specific *CDC6* PCR–product. Primers: random hexamers.

## Helicase loading and OCCM formation assays

Helicase loading was done as described in *Kang et al. (2014)*, except using 200 mM potassium glutamate (KGlut) instead of 300 mM. Briefly, 50 nM ORC, 100 nM Cdc6, 150 nM Mcm2–7/Cdt1 were combined in a 40 μL reaction containing 25 nM bead–bound 1.3 kB DNA including the *ARS1* origin. After mixing the proteins and DNA, reactions were shaken at 1,250 rpm at 25°C for 30 min (Thermomixer, Eppendorf), and then washed three times. For helicase–loading experiments, the three washes had buffer containing 300 mM K–Glut, 500 mM NaCl, and 300 mM KGlut respectively. DNA–bound proteins were eluted using DNase, run on SDS–PAGE gels, and detected using Krypton fluorescent stain (Fisher, PI–46629). To monitor OCCM formation, 5 mM ATPγS was used in place of ATP, and only washed with buffers containing 300 mM KGlut. Ime2 or CDK (150 nM or buffer control) was pre–incubated with the four purified proteins for 45 min before adding the bead–bound

*ARS1* DNA. For experiments examining the effects of phosphorylating individual proteins, 150 nM kinase was pre–incubated with the indicated protein in the '+kinase' reactions, and the other three proteins were mock phosphorylated. After 1 hr, the kinase inhibitor (1–NA–PP1 for Ime2, Sic1 for CDK) was added to both the '+kinase' and '–kinase' reactions, and the kinase itself was then added to the '–kinase' reactions to control for any effects that didn't depend on kinase activity. Phosphorylated and mock phosphorylated proteins were then combined and added to *ARS1* DNA to start the helicase–loading assay.

## Protein purifications

Ime2 (strain yDP159) and Ime2–AS (strain yDP554) were purified from yeast strains containing the $P_{GAL}$–*IME2*$^{stable}$–*3xFLAG* construct, which expressed amino acids 1–404 of *IME2* fused to a 3xFLAG epitope at the C–terminus. The Ime2–AS protein contains a M146G mutation. Eight liters of yeast were grown in YEP–Glycerol to $OD_{600}$ = 1.0 and induced with 2% galactose for 5 hr. Cells were lysed using a freezer mill in Buffer H (25 mM Hepes [pH = 7.6], 5 mM magnesium acetate [MgAc], 1 mM EDTA, 1 mM EGTA, 10% glycerol) containing 1 M Sorbitol, 0.02% NP–40, 2 mM ATP, 0.5 M KCl, 1x cOmplete Protease Inhibitors (Roche), and PhosSTOP phosphatase inhibitors (Roche). The lysate was clarified by centrifugation at 150,000 x g. KCl concentration was adjusted to 300 mM and the lysate was clarified again by centrifugation at 25,000 x g. Lysate was incubated with 1 mL M2–resin (Sigma, A2220) and washed with Buffer H with 300 mM KGlut and 0.01% NP–40 before elution with 3xFLAG peptide. Eluted protein was concentrated using a spin column (10 kDa cutoff, Vivaspin) and injected onto a Superdex 75 column (GE Healthcare) equilibrated with the same buffer. Peak Ime2–containing fractions were pooled and aliquoted.

Cdt1 (plasmid pALS16) was purified from *E. coli* Rosetta 2 cells induced overnight at 18°C. Cells were resuspended in 1x PBS, 10% glycerol, 1 mM DTT, and 300 mM NaCl, and lysed with lysozyme and sonication. The lysate was clarified by centrifugation at 150,000 x g. GST–Cdt1 was incubated with glutathione resin (Fisher Scientific, 17–5132–01) and washed with 50 mM Tris [pH7.6], 300 mM NaCl, 0.05% NP–40, 1 mM DTT, 1 mM EDTA, and 10% glycerol. Cdt1 was cleaved off the resin with Prescission Protease.

Clb5–CDK (ySK119) was purified as previously described (*Lõoke et al., 2017*). Cdc6 (pSKM033), ORC (ySDORC), Mcm2–7 (yST135), and Mcm2–7–Cdt1 (yST144) were purified as previously described (*Kang et al., 2014*).

## In vivo kinase inhibition

For strains containing *cdk1–as*, the kinase was inhibited with 10 μM 1–NM–PP1 (Toronto Research Chemicals, A603003). For strains containing *ime2–as*, the kinase was inhibited with 20 μM 1–NA–PP1 (Cayman Chemical Co., NC1049860). For experiments comparing the effects of inhibiting one or both kinases, both inhibitors were added to each strain regardless of genotype to normalize for the effect of the inhibitor without the corresponding analog–sensitive allele. Both inhibitors were prepared from 20 mM stocks in DMSO.

## ChIP–qPCR

Chromatin immunoprecipitations were performed as described with minor modifications (*Blitzblau et al., 2012*). 10 mL of cells were harvested for each sample, and 4% of the lysate was removed as an input control after sonication. Mcm2–7 was immunoprecipitated with 1.5 μL of UM185 (rabbit polyclonal antibody), whereas ORC was immunoprecipitated with 1 μL HM1108 (rabbit polyclonal antibody). Input and immunoprecipitated DNA from each sample were run in triplicate on a Light Cycler 480 II Real–Time PCR system (Roche). The relative amount of immunoprecipitated DNA vs. input DNA was calculated, and the highest sample, or the average of all G1 samples in *Figure 5*, was normalized to 1.0 within each experiment. Error bars represent the standard deviation from three PCR replicates.

## In vitro kinase assay

Ime2 was incubated with 100–200 nM of purified substrate protein (Cdc6, ORC, or Mcm2–7/Cdt1) in Buffer H + 200 mM K–Glut, 1 mM ATP, and 5 μCi [γ–$^{32}$P] ATP. Reactions were terminated after 45

min by boiling in sample buffer. Samples were loaded onto an SDS–PAGE gel, and total protein was visualized by Krypton fluorescent stain. $^{32}$P–modified proteins were detected by autoradiography.

## iTRAQ LC–MS/MS

All samples were analyzed with two biological replicates. 150 nM Ime2 (or buffer control) was incubated with 400 nM Cdc6 under the same buffer conditions and for the same time as in the helicase–loading assay. Proteins were reduced, alkylated, and digested with trypsin. Peptides were labeled using 4 of the 6 channels from the TMT 6plex kit (Thermo) performed per manufacturer's instructions. Samples labeled with the four different isotopic TMT reagents were combined and concentrated to completion in a vacuum centrifuge. house, 6 cm of 10 μm C18) and a self–pack 5 μm tip analytical column (12 cm of 5 μm C18, New Objective) over a 140 min gradient before nanoelectrospray using a QExactive Plus mass spectrometer (Thermo). The parameters for the full scan MS were: resolution of 70,000 across 350–2000 $m/z$, AGC 3e$^6$, and maximum IT 50 ms. The full MS scan was followed by MS/MS for the top 10 precursor ions in each cycle with a NCE of 28 and dynamic exclusion of 30 s. Raw mass spectral data files (.raw) were searched using Proteome Discoverer (Thermo) and Mascot version 2.4.1 (Matrix Science). Mascot search parameters were: 10 ppm mass tolerance for precursor ions; 15 mmu for fragment ion mass tolerance; 2 missed cleavages of trypsin; fixed modification were carbamidomethylation of cysteine and TMT 6plex modification of lysines and peptide N–termini; variable modifications were methionine oxidation, tyrosine phosphorylation, and serine/threonine phosphorylation. TMT quantification was obtained using Proteome Discoverer and isotopically corrected per manufacturer's instructions, and were normalized to the mean of each TMT channel. Only peptides with a Mascot score greater than or equal to 25 and an isolation interference less than or equal to 30 were included in the data analysis. Mascot peptide identifications, phosphorylation site assignments, and quantification were verified manually with the assistance of CAMV (*Curran et al., 2013*). Phosphorylation sites were assigned based on having an average enrichment of >4–fold in the Ime2–treated samples compared to control samples.

## Acknowledgements

We thank Angelika Amon, Iain Cheeseman, Audra Amasino, Ishara Azmi, Caitlin Blank, and Annie Zhang for comments on the manuscript. We thank all members of the Bell laboratory for helpful discussions. We thank Angelika Amon and John Diffley for yeast strains. D.V.P. was supported in part by a NIH Pre-Doctoral Training Grant (GM007287). SPB is an investigator with the Howard Hughes Medical Institute. This work was supported in part by the Koch Institute Support Grant P30-CA14051 from the NCI. We thank the Koch Institute Swanson Biotechnology Center for technical support, specifically the Biopolymers core.

## Additional information

### Funding

| Funder | Grant reference number | Author |
|---|---|---|
| Howard Hughes Medical Institute | Investigator Award | Stephen P Bell |
| National Cancer Institute | Biopolymer Facility Support | Stephen P Bell |
| National Institute of General Medical Sciences | Gradaute Student Fellowship | David V Phizicky |

The funders had no role in study design, data collection and interpretation, or the decision to submit the work for publication.

### Author contributions

David V Phizicky, Conceptualization, Data curation, Formal analysis, Investigation, Visualization, Methodology, Writing—original draft; Luke E Berchowitz, Supervision, Methodology, Writing—review and editing; Stephen P Bell, Conceptualization, Supervision, Funding acquisition, Project administration, Writing—review and editing

## Author ORCIDs
David V Phizicky ⓘ http://orcid.org/0000-0002-4219-702X
Stephen P Bell ⓘ http://orcid.org/0000-0002-2876-610X

## Decision letter and Author response
Decision letter https://doi.org/10.7554/eLife.33309.031
Author response https://doi.org/10.7554/eLife.33309.032

## Additional files

### Supplementary files
• Supplementary file 1. In vitro Ime2–phosphorylation sites on Cdc6 from iTRAQ LC–MS/MS. Related to *Figure 5E*. Shown are the phosphorylated Cdc6 peptides, the specific phosphorylated residue(s), the relative amount of those phosphopeptides detected in both biological replicates of buffer–treated and Ime2–treated Cdc6, and the average enrichment upon Ime2–treatment. See Materials and Methods for iTRAQ LC–MS/MS details.
DOI: https://doi.org/10.7554/eLife.33309.027

• Transparent reporting form
DOI: https://doi.org/10.7554/eLife.33309.028

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
