## [Decision Letter]

Thank you for submitting your article "Multiple kinases inhibit origin licensing and helicase activation to ensure reductive cell division during meiosis" for consideration by *eLife.* Your article has been favorably evaluated by Andrea Musacchio (Senior Editor) and three reviewers, one of whom, Bruce Stillman (Reviewer #1), is a member of our Board of Reviewing Editors.

The reviewers have discussed the reviews with one another and the Reviewing Editor has drafted this decision to help you prepare a revised submission.

Summary:

This paper addresses a fundamental, unresolved question concerning the regulation of DNA replication during progression through meiosis: How do cells carry out two rounds of chromosome segregation without an intervening round of DNA replication? The authors elucidate several distinct mechanisms which inhibit replication initiation between two consecutive meiotic nuclear divisions (MI and MII). In mitotically dividing cells, oscillation of CDK ensures that replication origins fire once per cell cycle: MCM helicase loading occurs when the cellular CDK level is low, and MCM is activated while loading is prevented when CDK level is high. In meiosis, the CDK level drops once at the MI to MII transition, but replication initiation does not occur. In frog oocyte extracts, addition of a CDK inhibitor promotes S phase between MI and MII, and in yeast simultaneous inhibition of CDK and Ime2 (a meiosis specific kinase related to CDK) leads to nuclear accumulation of an MCM protein during nuclear division stages. Thus, it is suggested that the two kinases function to prevent replication initiation when MCM associates to replication origins. But it was not known precisely how these two kinases do so.

The authors employed a previously developed system to synchronize meiosis progression in which they use a yeast strain with inducible Ndt80 (a transcription factor that triggers exit from meiotic prophase). This strain arrests in meiotic prophase and resumes progression upon Ndt80 induction. Using this system, authors found that MCM ChIP signals at replication origins are decreased after the release from prophase arrest and that either CDK or Ime2 activity is required to keep MCM ChIP signal at a low level. Results of an in vitro MCM loading assay demonstrate that Ime2 and CDK prevent MCM loading by phosphorylating both overlapping and distinct targets. Furthermore, during nuclear meiotic division stages, CDK and Ime2 repress Cdc6 (a protein that recruits MCM to origins) in part by phosphorylating Cdc6 within phospho-degron domains, and Sld2 (an MCM activator) degradation is dependent on phosphorylation sites for CDK and Cdc5 (the polo-like kinase).

This elegant paper provides detailed mechanistic insight into how replication is inhibited at the MI-MII transition stage in meiosis. The data are excellent and the paper is very well written. The paper is highly appropriate for *eLife*. However, there are a few areas that should be addressed to strengthen the paper.

Minor points:

1) The dephosphorylation of Orc2 and Orc6 are very subtle and it is not clear if there is "a partial but detectable reversal of ORC inhibition" since the authors do not know if partial dephosphorylation is sufficient for reversing inhibition of ORC function. This statement should be qualified.

2) Figure 3. The amount of Ime1 kinase that inhibits Mcm207 loading is near stoichiometric with the pre-RC assembly proteins. Similarly, the kinase assay in Figure 3—figure supplement 2 suggests stoichiometric phosphorylation. These data might suggest that like CDK, Ime2 binds one or more of the pre-RC proteins. Has this been tested?

3) A rather obvious question is whether DNA re-replication does in fact occur when the mechanisms delineated here are compromised. Have the authors tried to see if replication occurs when CDK and Ime2 are artificially oscillated by inhibition followed by washing out of the inhibitors in the non-phosphorylatable sld2 mutant background? The discussion point about Cdc7-Dbf4 regulation raises the possibility of yet other mechanisms to inhibit re-replication, but it would strengthen the paper to test just how redundant these mechanisms are.

4) Please show representative micrographs to illustrate the cell cycle staging. This could be provided in Figure 1—figure supplement 1 (would not be needed for all).

5) Figure 1 and C. No negative control is presented to evaluate ChIP signal specificity. This is more important for the ORC ChIP, but would have been good to show for MCM as well. Also, it would be better if the y-axis in this and other ChIP plots showed efficiency (% of input), rather than the max-normalized value (arbitrary units).

6) Figure 5—figure supplement 2. Experiments here and elsewhere were conducted in an artificially synchronized meiosis using inducible Ndt80. Normally, the pachytene checkpoint monitors failures in homologous recombination and chromosome synapsis, and inhibits NDT80 expression until the checkpoint is satisfied. But even in wild-type meiosis, normal recombination processes delay Ndt80 activation to prevent premature meiotic entry. Thus, it is important to consider that the artificial synchrony system allows for cells to be driven out of prophase before they have completed recombination and other events. Normally, the majority of cells have completed prophase by 6 hr after the induction of sporulation. However, the sporulation is intrinsically variable. In the case of cells that are progressing somewhat slowly, "premature" induction of Ndt80 could cause a substantial number of achiasmate chromosomes, which would lead to activation of the spindle assembly checkpoint. This could be problematic especially when the next manipulations take place within a fixed time interval. Such a scenario perhaps explains why a lower fraction of cells completed MI in Figure 5—figure supplement 2. If possible, it might be useful to supplement or replace these data with better cultures or with cultures synchronized with an alternative method (e.g., pCUP1-IME4/IME4 strain or cdc7-as inhibition followed by inhibitor washout). At the least, it is not appropriate to claim the 7h45' time point as MI-MII, since nearly half of the cells are still in metaphase I, so the text should be modified accordingly.

7) Figure 6 needs to be bolstered with additional data. The western signals need to be normalized to input control, for which Ponceau staining is probably sub-optimal. A dilution series to demonstrate linearity of the western blot quantification is needed (can be supplemental). There are no error bars on this experiment; how many times was this repeated?

---

## [Author Response]

[…] This elegant paper provides detailed mechanistic insight into how replication is inhibited at the MI-MII transition stage in meiosis. The data are excellent and the paper is very well written. The paper is highly appropriate for eLife. However, there are a few areas that should be addressed to strengthen the paper.Minor points:1) The dephosphorylation of Orc2 and Orc6 are very subtle and it is not clear if there is "a partial but detectable reversal of ORC inhibition" since the authors do not know if partial dephosphorylation is sufficient for reversing inhibition of ORC function. This statement should be qualified.

We understand and agree with the reviewers’ concern with this statement. We have now qualified our statement to read “a partial but detectable decrease of ORC phosphorylation”.

2) Figure 3. The amount of Ime1 kinase that inhibits Mcm207 loading is near stoichiometric with the pre-RC assembly proteins. Similarly, the kinase assay in Figure 3—figure supplement 2 suggests stoichiometric phosphorylation. These data might suggest that like CDK, Ime2 binds one or more of the pre-RC proteins. Has this been tested?

The most direct test for Ime2 binding to pre-RC proteins we have done is to look for the association of Ime2 with the ORC-Cdc6-Cdt1-MCM complex formed in ATPyS. Because this is a stable complex containing all the components required for helicase loading we felt this was the best test for Ime2 binding. In contrast to a model in which Ime2 inhibits helicase loading through binding to helicase-loading proteins, we found that Ime2 did not stoichiometrically bind to these proteins in this experimental condition (see Author response image 1). We realize that this does not exclude binding of Ime2 to these proteins at other times in the reaction. On the other hand, we show that Ime2-mediated inhibition of helicase loading is completely dependent on its kinase activity (Figure 3), indicating that any potential role for Ime2-binding to its substrates is not sufficient for inhibition.

**Author response image 1. respfig1:** OCCM Formation Assay.

3) A rather obvious question is whether DNA re-replication does in fact occur when the mechanisms delineated here are compromised. Have the authors tried to see if replication occurs when CDK and Ime2 are artificially oscillated by inhibition followed by washing out of the inhibitors in the non-phosphorylatable sld2 mutant background? The discussion point about Cdc7-Dbf4 regulation raises the possibility of yet other mechanisms to inhibit re-replication, but it would strengthen the paper to test just how redundant these mechanisms are.

We agree with the reviewers that this would be a very exciting experiment. Unfortunately, we found that inhibition of Cdk1-as during meiosis followed by washing away the inhibitor does not allow Cdk1-as to be reactivated. These results are consistent with previous experiments showing that transient inhibition of Cdk1-as at any point during meiosis caused cells to be unable to progress any further in meiosis (Holt et al., 2007 – see Discussion). Like Holt et al., we believe that these data suggest a requirement for basal CDK activity during meiosis. It is possible that complete CDK inhibition allows for Sic1 to accumulate preventing CDK reactivation upon inhibitor washout.

4) Please show representative micrographs to illustrate the cell cycle staging. This could be provided in Figure 1—figure supplement 1 (would not be needed for all).

We have now included representative images of cells in Metaphase I, Anaphase I, Metaphase II, and Anaphase II in Figure 1—figure supplement 1.

5) Figure 1 and C. No negative control is presented to evaluate ChIP signal specificity. This is more important for the ORC ChIP, but would have been good to show for MCM as well. Also, it would be better if the y-axis in this and other ChIP plots showed efficiency (% of input), rather than the max-normalized value (arbitrary units).

We agree with the reviewers that reporting the negative control and% of input is important. We have now included an additional figure (Figure 1—figure supplement 2) showing that our ChIP for both ORC and MCM is specific to origin DNA compared to non-origin DNA. The previous “Figure 1—figure supplement 2” is now “Figure 1—figure supplement 3”.

We have also included the % of input detected for each protein and origin combination in the corresponding figure legends. We have not changed the y-axes of our graphs, however, to allow protein association at different origins to be compared on the same graph (the peak% input value varies significantly between origins).

6) Figure 5—figure supplement 2. Experiments here and elsewhere were conducted in an artificially synchronized meiosis using inducible Ndt80. Normally, the pachytene checkpoint monitors failures in homologous recombination and chromosome synapsis, and inhibits NDT80 expression until the checkpoint is satisfied. But even in wild-type meiosis, normal recombination processes delay Ndt80 activation to prevent premature meiotic entry. Thus, it is important to consider that the artificial synchrony system allows for cells to be driven out of prophase before they have completed recombination and other events. Normally, the majority of cells have completed prophase by 6 hr after the induction of sporulation. However, the sporulation is intrinsically variable. In the case of cells that are progressing somewhat slowly, "premature" induction of Ndt80 could cause a substantial number of achiasmate chromosomes, which would lead to activation of the spindle assembly checkpoint. This could be problematic especially when the next manipulations take place within a fixed time interval. Such a scenario perhaps explains why a lower fraction of cells completed MI in Figure 5—figure supplement 2. If possible, it might be useful to supplement or replace these data with better cultures or with cultures synchronized with an alternative method (e.g., pCUP1-IME4/IME4 strain or cdc7-as inhibition followed by inhibitor washout). At the least, it is not appropriate to claim the 7h45' time point as MI-MII, since nearly half of the cells are still in metaphase I, so the text should be modified accordingly.

We have altered the text (subsection “CDK and Ime2 cooperate to inhibit Mcm2-7 loading and Cdc6 expression during the MI-MII transition”) and Figure 5 to more accurately reflect the stage of meiosis upon CDK/Ime2 inhibition. We specifically highlight the percentage of cells in anaphase I and acknowledge in the text that not all cells are at this stage of meiosis (first paragraph of the aforementioned subsection). We focus on these cells because anaphase I is good marker for cells as CDK activity decreases upon entry into the MI-MII transition. Two pieces of evidence support this claim. First, previous work demonstrated that Cdk1-cyclin B complexes have decreased activity in anaphase I relative to metaphase I (Carlile and Amon, 2008). Second, the peak of anaphase I corresponds with the decreased ORC phosphorylation and the beginning of Cdc6 reaccumulation we observe in Figure 2, markers of decreased CDK activity. We note that it is likely that CDK, and potentially Ime2, inhibits helicase loading throughout the meiotic divisions and not just at the MIMII transition. Thus, the fact that we see helicase reloading at any time during the meiotic divisions (there is little doubt that the majority of the cells have at least entered MI) strongly supports our primary conclusion: that the two kinases both contribute to inhibiting helicase loading.

The order of Figure 5—figure supplement 1 and 2 have been switched from the previously submitted version to account for changes in the text.

With regard to why fewer cells have completed MI in the current Figure 5—figure supplement 1 and 1D relative to 1A and 1C of the same figure, this difference is because CDK has been inhibited in the 1B and 1D cultures but not in the other two cultures. CDK is required for progression through MI, and thus its inhibition prevents MI completion in a large proportion of cells. We have now included immunofluorescence quantification from an 8h30min time point from these same cultures that was not treated with the kinase inhibitors (Figure 5—figure supplement 1). This data shows that >85% of cells have completed MI at this time point in all four cultures.

7) Figure 6 needs to be bolstered with additional data. The western signals need to be normalized to input control, for which Ponceau staining is probably sub-optimal. A dilution series to demonstrate linearity of the western blot quantification is needed (can be supplemental). There are no error bars on this experiment; how many times was this repeated?

We understand the reviewers’ concern with the data in this form. We generally prefer Ponceau staining over an individual protein since monitoring many proteins as opposed to just PGK1 make the normalization less dependent on the stable expression of one protein. However, we do understand that it is more difficult to quantify the Ponceau staining. Thus, we have now normalized our western signals to a PGK1 loading control and found the same results. This experiment has now been repeated three times, and we report the mean and standard deviation from these three experiments. The text has been modified to reflect the quantification shown in Figure 6 (subsection “Cdc5 and CDK promote the degradation of Sld2, an essential helicase- activation protein”, last paragraph). We have also included a dilution series to show that the western blot quantification is accurate within a factor of two across a 32-fold dilution range (Figure 6—figure supplement 3).